# Exploring Sea Lice Vaccines against Early Stages of Infestation in Atlantic Salmon (*Salmo salar*)

**DOI:** 10.3390/vaccines10071063

**Published:** 2022-07-01

**Authors:** Antonio Casuso, Valentina Valenzuela-Muñoz, Bárbara P. Benavente, Diego Valenzuela-Miranda, Cristian Gallardo-Escárate

**Affiliations:** 1Interdisciplinary Center for Aquaculture Research (INCAR), University of Concepción, Concepción 4030000, Chile; acasuso@udec.cl (A.C.); valevalenzuela@udec.cl (V.V.-M.); bbenavente@udec.cl (B.P.B.); divalenzuela@udec.cl (D.V.-M.); 2Laboratory of Biotechnology and Aquatic Genomics, Department of Oceanography, University of Concepción, Concepción 4030000, Chile

**Keywords:** *Salmo salar*, *Caligus rogercresseyi*, transcriptome, ectoparasite, vaccine

## Abstract

The sea louse *Caligus rogercresseyi* genome has opened the opportunity to apply the reverse vaccinology strategy for identifying antigens with potential effects on lice development and its application in sea lice control. This study aimed to explore the efficacy of three sea lice vaccines against the early stage of infestation, assessing the transcriptome modulation of immunized Atlantic salmon. Therein, three experimental groups of *Salmo salar* (Atlantic salmon) were vaccinated with the recombinant proteins: Peritrophin (prototype A), Cathepsin (prototype B), and the mix of them (prototype C), respectively. Sea lice infestation was evaluated during chalimus I-II, the early-infective stages attached at 7-days post infestation. In parallel, head kidney and skin tissue samples were taken for mRNA Illumina sequencing. Relative expression analyses of genes were conducted to identify immune responses, iron transport, and stress responses associated with the tested vaccines during the early stages of sea lice infection. The vaccine prototypes A, B, and C reduced the parasite burden by 24, 44, and 52% compared with the control group. In addition, the RNA-Seq analysis exhibited a prototype-dependent transcriptome modulation. The high expression differences were observed in genes associated with metal ion binding, molecular processes, and energy production. The findings suggest a balance between the host’s inflammatory response and metabolic process in vaccinated fish, increasing their transcriptional activity, which can alter the early host–parasite interactions. This study uncovers molecular responses produced by three vaccine prototypes at the early stages of infestation, providing new knowledge for sea lice control in the salmon aquaculture.

## 1. Introduction

The aquaculture industry’s interest in disease prevention methods has increased its focus on developing novel and effective fish vaccines [1]. Nowadays, many vaccines are used for bacterial and viral control [2,3,4]. However, vaccines for fish ectoparasites remain the major challenge in aquaculture. The scientific knowledge of the host–parasite interactions is essential for developing ectoparasite vaccines [5,6]. Therein, omics approaches can expand our understanding of the molecular mechanism involved in parasite biology and the interplaying with the host. Here, the information achieved can be used as new targets for parasite control [6,7]. In this context, the vaccine design based on reverse vaccinology represents a valuable approach for antigens discovery. Notably, the increasing bioinformatic analysis capacity improves the screening process of biomolecules potentially effective as vaccine candidates against fish parasites [8,9].

Two proteins with antigenic potential for developing a vaccine to control ectoparasites are cathepsin and peritrophin [10,11]. The role of both proteins in essential pathways for the development of host–parasite interaction makes them good candidates. Invertebrate cathepsins are proteases with many physiological functions [12], such as degradation and turnover of intracellular proteins and immune response [13,14]. Additionally, cathepsins participate in hemoglobin digestion in hematophagous parasites, such as ticks and hookworms [14,15,16]. Moreover, in some hematophagous parasites, such as sea lice, cathepsins have been identified in all developmental stages and are related to host–pathogen interaction mechanisms [17,18,19]. Similarly, peritrophin proteins are involved in several physiological processes. These proteins are part of the peritrophic matrix that lines the intestine of several invertebrates [20,21]. Peritrophins participate in pathogen infection control [22,23] and intestinal protection during host–pathogen interaction [24,25]. For example, a peritrophin from the *Anopheles gambiae* mosquito takes heme groups derived from blood degradation, preventing the formation of free radicals and protecting the mosquito gut from free heme toxicity [26,27]. The significant functional role of both proteins has stimulated the evaluation of their antigenic potential to control pests using cathepsins [11,28,29,30] and peritrophins [10,31,32].

Different ectoparasite vaccine prototypes have been tested for sea lice control. For instance, the antigens my32, TT-P0, potassium chloride transporter, and amino acid transporter have previously been evaluated [33,34,35]. These studies report parasite burden reduction in *S. salar*, with efficacy between 30 to 57%. Furthermore, in response to ectoparasite vaccination, fish–host humoral and cellular components have been observed [36,37,38]. Notably, has been observed the expression modulation of IgM and IgT, antioxidant response, and related inflammatory genes in fish immunized and exposed to the ectoparasite infestation [37,39,40]. Moreover, the use of antigens associated with the host–iron metabolism modulation was recently reported [41]. Our recently published study demonstrated that the IPath^®^ vaccine, with chelating iron function, reported a sea lice burden reduction of over 90% [41]. The findings suggested that identified antigens in marine ectoparasites trigger host humoral and cellular immune responses and activate host nutritional immunity during lice infestations [42].

In Chile, the sea louse, *Caligus rogercresseyi* is considered one of the most significant economic impacts on the salmon aquaculture industry [43,44]. This copepod species feeds from fish mucus, skin, and blood, producing lesions, weight loss, and immunosuppression in fish [42,45,46]. In *C. rogercresseyi*, the host–parasite interactions begin when the copepodite stage recognizes the host, developing the frontal filament for fish attachment on the skin [47,48]. Subsequently, sea lice begin to feed as chalimus I [49], the stage where species-specific differences have been reported, and different lice infestation success has been reported among resistant and susceptible hosts [50,51]. In recent years, many studies have provided information to improve the understanding of the *C. rogercresseyi* infection mechanisms and their host responses [42,52,53]. Notably, the *C. rogercresseyi* genome has been sequenced and assembled in pseudo-chromosomes [53], providing pivotal molecular information to uncover relevant genetic components and biological processes during the life cycle of *C. rogercresseyi*. Herein, modulation of genes associated with immune response, iron transport, and stress response have been identified in Atlantic salmon during the early stages of infection. In parallel, bioinformatic analyses have identified specific proteins with essential roles during the early-developmental stages of copepodid and chalimus. Taken from these previous findings, we aimed to evaluate three vaccine prototypes in Atlantic salmon against sea lice infection. The vaccine assessment was conducted through transcriptome profiling of immunized fish and the efficacy level during early *C. rogercresseyi* infection.

## 2. Materials and Methods

### 2.1. Antigens Selection and Purification of Recombinant Proteins

From *C. rogercresseyi* genome database [53], a Blastx analysis was performed using reference sequences previously reported to secretory/excretory proteins (SEPs) by Hamilton et al. [54]. According to the authors, 187 individual proteins were identified in the SEPs collected from the sea lice *Lepeophtheirus salmonis*. Putative proteins differentially expressed during copepodid and chalimus I–II stages were also included in the analysis. The *C. rogercresseyi* transcriptome evaluated the SEPs profiling during the ontogeny. Using a transcriptome database of *C. rogercresseyi* ontogeny [55] an RNA-Seq analysis was performed in CLC Genomic Workbench software v21 (Qiagen Bioinformatics, USA). The settings parameters were a minimum length fraction = 0.6 and a minimum similarity fraction (long reads) = 0.5. The expression value was set as Transcripts Per Kilobase Million (TPM). Gene expression changes were compared with Kal’s test statistical analysis (*p* = 0.0005; FDR corrected). Furthermore, the whole *C. rogercresseyi* genome modulation was explored using the Chromosome Genome Expression (CGE) index according to Valenzuela-Muñoz et al. [56]. From this analysis two genes, cathepsin and peritrophin were selected because their expression profile along the sea lice ontogeny. Both genes were characterized using Geneious Pro software (Version 11.0.9, Biomatters Ltd.a., Auckland, NI, New Zealand).

The two selected nucleotide sequences were synthesized and cloned into the vector pET-30a (+) by GenScript (Piscataway, NJ, USA). Subsequently, the vectors were transformed into *Escherichia coli* BL21 strains, grown in LB culture medium at 37 °C until reaching an approximated optical density of 0.6 (OD_600_ ≈ 0.6). Protein expression was induced by adding 0.5 mM isopropyl b-D-1-0.5 mM thiogalactopyranoside (IPTG) (Invitrogen, ThermoFisher Scientific, Waltham, MA, USA). Bacterial extracts were sonicated (95% amplitude, 5 s ON/10 s OFF, 10 min), and centrifugation separated phases. Pellet were solubilized for 2 h in constant motion (20 mM Tris; 10 mM Imidazole, 3 M Urea; 5 mM DTT, pH 12). Peritrophin and cathepsin were purified on AKTAPrime Plus (GE Healthcare, Chicago, IL, USA) liquid chromatography using a HisTrap™ FF column (5 mL) (GE Healthcare). In the case of PER, the fractions were further purified on a HiLoad™ 16/600 Superdex™ 75 column (GE Healthcare). The purified proteins were dialyzed against PBS (phosphate-buffered saline) with a SnakeSkin™ dialysis tube (Thermo Scientific) with a molecular weight limit of 3.5 K. Protein identification was verified in each purification step by SDS-PAGE (12%) and by Western blot using an anti-6xHis antibody HRP-conjugate (Thermo Fisher Scientific) in 1:2000 dilution and revealed using ECL™ Western Blotting System (Thermo Fisher Scientific).

### 2.2. Fish Vaccination, Challenge Trial, and Ethics Statement

Vaccine prototypes were formulated in 100 µL per dose in a ratio of 30% antigen/70% adjuvant, using 30 µg of antigen and the commercial adjuvant Montanide™ ISA 761 VG. Three prototype vaccines were prepared: prototype A, recombinant peritrophin; prototype B, recombinant cathepsin; and prototype C, the combination of peritrophin and recombinant cathepsin (50/50%) (Appendix A). PBS plus adjutant was also formulated as a control vaccine group. *S. salar* of 100 gr were acclimatized for two weeks in the experimental laboratory of the Marine Biological Station, University of Concepción, Dichato, Chile. Fish were injected intraperitoneally in triplicate and divided into four experimental groups with 20 fish per tank. The groups were immunized with the vaccine prototypes A, B, and C, and the control group. After 400 UTAs, fish were infested with 35 copepodites per fish. At 7 days post-infestation (dpi), samples of fish skin and head kidney were taken, fixed in RNA Stabilization Reagent^®^ (Ambion, Life Technologies™, Carlsbad, CA, USA), and stored at −80 °C until RNA extraction. Further, sea lice count on the fish surface was recorded.

The animal protocol developed in this research was approved by the Ethics, Bioethics, and Biosafety Committee of the Vice-Rectory for Research and Development of the University of Concepción, Chile. This research was carried out following the recommendations of the International Guiding Principles for Biomedical Research Involving Animals (Council for International Organization of Medical Science and The International Council for Laboratory Animal Science, 2012), the document “Bioethical Aspects of Animal Experimentation”, edited by the Bioethics Advisory Committee of FONDECYT-CONICYT (2009) and the “Biosafety Standards and Associated Risks”, version 2018, edited by FONDECYT-CONICYT.

### 2.3. High-Throughput Transcriptome Sequencing

Total RNA of head kidney and skin samples were extracted using the Trizol Reagent (Ambion, USA) following the manufacturer’s instructions. The isolated RNA’s quality, purity, and quantity were measured in the TapeStation 2200 (Agilent Technologies Inc., Santa Clara, CA, USA). Samples with a RIN over 8.0 were five individuals from each tissue, and conditions were pooled for TrueSeq Stranded mRNA Illumina library synthesis. The transcriptome sequencing was performed on the Hi-Seq Illumina platform by the Macrogen Inc. (Seoul, Korea). Raw data were trimmed in CLC Genomics Workbench (Version 21.0.3). De novo assemblies were performed using data sets from each tissue and treatment, with an overlap criterion of 70% and a similarity of 0.9 to exclude paralogous sequence variants (Renaut, Nolte, and Bernatchez 2010). Contigs were annotated by BlastX analysis using a database constructed from GenBank and UniprotKB/Swiss-Prot, with a cutoff E-value of 1E-5.

### 2.4. RNA-Seq Analysis

The obtained contigs were used as a reference for RNA-seq analyses using the CLC Genomic Workbench software (Version 21.0.3). Atlantic salmon head kidney and skin sequencing raw data were separately mapped against all annotated contigs for each experimental group. The settings were a minimum length fraction = 0.8 and a minimum similarity fraction (long reads) = 0.8. Expression value was established as transcripts per million reads (TPM). The distance metric was calculated with the Manhattan method, and a Kal’s statistical analysis test was used to compare gene expression levels in fold change (*p* = 0.0005; FDR corrected). Finally, differential expression analysis was separately conducted for tissue samples using as references the transcripts expressed in the control group, injected with PBS. Expression values were plotted on a heat map using the differentially expressed contigs as a statistical comparison (FDR *p*-value < 0.05). Transcripts differently expressed were represented in a Venn diagram using contigs with absolute fold-change values ≥ 4 and *p*-value < 0.05.

### 2.5. Gene Ontology and Pathway Enrichment Analyses

Gene Ontology (GO) enrichment analyzes were performed using the Blast2GO plug-in of the CLC Genome software, with predetermined parameters. Transcripts shared among the three prototypes and transcripts differentially expressed in head kidney, and skin transcriptomes were subjected to the Fisher’s Exact Test. GO analyzes were performed to identify the biological processes (BP) and molecular functions (MF) most represented in each experimental group. In addition, contigs differently expressed were analyzed for orthologous assignment, and pathway mapping was performed using the KEGG Automatic Annotation Server (KAAS) database [57]. The genomes of *S. salar* were used as references.

### 2.6. RNA Extraction and RT-qPCR

The evaluation of vaccine prototypes against early sea louse infestation was conducted by seven immune response genes and seven iron regulation genes (Appendix A) in the skin and head kidney by RT-qPCR, according to Valenzuela-Muñoz et al. [41]. Additionally, the reaction efficiency of each primer was calculated (Appendix A). The same pool RNA used for the high-throughput transcriptome was used for RT-qPCR. The RT-qPCR standardization was carried out according to the MIQE guidelines [58]. cDNA was synthesized starting at 200 ng/µL of initial total RNA and using the RevertAid H Minus First Strand cDNA Synthesis kit (Thermo Fisher Scientific), following the manufacturer’s directions. The RT-qPCR reaction was performed on the QuantStudio™ 3 Real-Time PCR System (Applied Biosystems). The comparative ΔΔCt method was used for gene expression quantification [59]. The selection of the reference gene for the experiment was based on evaluating the stability of the elongation factor-α, b-tubulin, and 18S genes by NormFinder. Through this, the elongation factor-α was selected and data were normalized using this gene as reference. The PowerUp™ SYBR™ Green Master Mix (Thermo Fisher Scientific, Waltham, MA, USA) was used in a final reaction volume of 10 µL. Amplification was carried out under the following conditions: 95 °C for 10 min, 40 cycles of 95 °C for 15 s, and alignment temperature for 30 s (Appendix A), followed by 30 s at 72 °C. TaqMan probes previously described for *S. salar* were used to measure IgM and IgT expression levels [38]. Each reaction was carried out in a final volume of 12 µL using the commercial Kapa Probe Fast Universal qPCR kit (Kapa Biosystems, Potters Bar, HRT, UK). The amplification reaction was performed on a QuantStudio™ 3 Real-Time PCR System (Applied Biosystems^®^, Life Technologies, Foster City, CA, USA), under the following conditions: 95 °C for 10 min, 45 cycles at 95 °C for 30 s, and 60 °C for 1 min. Statistical analyzes were performed in the GraphPad Prism 6.0 software (San Diego, CA, USA). The Shapiro–Wilk test for statistical analysis determined the distribution of data. Additionally, the data were evaluated by one-way ANOVA and Tukey’s post hoc analysis determined significant differences. Statistically significant values were set at *p* < 0.05.

## 3. Results

### 3.1. Identification of Excretory/Secretory Proteins (SEPs) during the Early-Developmental Stage of Sea Lice

From the *C. rogercresseyi* genome database, a Blastx analysis was performed using reference sequences previously reported to secretory/excretory proteins (SEPs) by Hamilton et al. [54]. According to the authors, 187 individual proteins were identified in the SEPs collected from the sea lice *Lepeophtheirus salmonis*. Proteins differentially expressed between copepodid and chalimus I-II stages were also included. The *C. rogercresseyi* transcriptome evaluated the SEPs profiling during the ontogeny. In addition, the whole *C. rogercresseyi* genome modulation was explored using the Chromosome Genome Expression (CGE) index according to Valenzuela-Muñoz et al. [56]. This approach helps scan differentially expressed chromosome regions from data sets with several experimental conditions. Here, the transcriptomes of nauplius, copepodid, chalimus I-II, chalimus III-IV, and male and female stages were compared, showing specific gene clusters annotated on specific chromosome regions associated with peritrophin and cathepsin proteins. For instance, differentially expressed Chromosome regions during sea lice lifecycle were localized in the Chr2, Chr9, Chr12, Chr13, Chr19, Chr20, and Chr21 (Figure 1).

### 3.2. Transcriptome Modulation in Head Kidney, and Skin

Head kidney tissue RNA-Seq analysis showed differences in expression profiles among the experimental groups after 7 dpi (Figure 2). Two expression clusters were identified from the RNA-seq analysis (Figure 2A). Cluster 1 was highly expressed in samples of fish vaccinated with prototypes B and C. The annotated genes were ovarian aromatase, XK-related protein 8, Cytochrome b, Ig mu chain C region membrane-bound, Centrosomal protein, Protein NLRC5, and Collectin-11 (Appendix A). Cluster 2 exhibited a high expression level in control group samples presenting genes such as Hemoglobin subunit alpha and beta, Cathepsin D, APC membrane recruitment protein, and Ferritin heavy subunit (Appendix A). In addition, differential expression analysis among vaccinated fish related to the control group showed 1060, 532, and 1290 transcripts exclusive expressed in prototypes A, B, and C, respectively (Figure 2B, Table 1). Furthermore, the three vaccine prototypes showed many transcripts upregulated, with fold change values < 10 (Figure 2C). Interestingly, head kidney samples of fish vaccinated with prototypes B and C, both with cathepsin in the formulation, showed a higher number of transcripts differently modulated than fish vaccinated with prototype A, which preset as antigen peritrophin protein.

RNA-Seq analysis of *S. salar* skin, exposed to the different vaccines, exhibited three main clusters (Figure 3A). Cluster 1, highly expressed in the control group, showed genes annotated as Troponin C, Calmodulin, Myosin heavy chain, and Tropomyosin alpha-1 chain (Appendix A). Interestingly, skin transcripts of fish vaccinated with prototype A were separately grouped in Cluster 2. In this cluster, the transcripts upregulated were observed Actin, Collagen alpha-2 (I) chain, Creatine kinase, and Myosin heavy chain. Furthermore, contigs of cluster 3 exhibited upregulation in salmons vaccinated with prototypes B and C. The annotation of this cluster showed genes such as Glyceraldehyde-3-phosphate dehydrogenase, Fructose-bisphosphate aldolase A, Myosin heavy chain, Calmodulin, and Interleukin-17 receptor D.

Moreover, differential expression analysis among the control group and vaccines prototype showed 7333, 3315, and 2739 transcripts exclusively expressed in salmon skin vaccinated with prototypes A, B, and C, respectively (Figure 3B). In addition, from the annotation of sharing contigs were annotated genes as lactosylceramide transferase and calcium ATPase (Table 2). Finally, a high number of upregulated transcripts were observed in skin samples from vaccinated fish (Figure 3C). At the difference in head kidney tissue, it was possible to observe transcripts with fold changes values up to 100.

### 3.3. GO Enrichment and KEGG Pathway Analysis

Transcripts differently expressed were annotated using GO annotation (Figure 4 and Figure 5). Fish vaccinated with prototype A showed fewer transcripts associated with a GO term compared with prototypes B and C. In head kidney tissue from fish vaccinated with prototype A exhibited among the biological process (BP) transcripts annotated as peptidyl-tyrosine phosphorylation, epithelial tube formation, peptidyl-serine modification, and response to a mechanical stimulus (Figure 4). Among the molecular functions (MF), transcription factor binding was the most enriched GO term. Regarding skin samples of fish vaccinated with prototype A the BPs annotated were developmental process, anatomical structure development, and multicellular organism development (Figure 5). In addition, the MFs identified were actin binding, structural molecule activity, and phosphatase activity. Regarding fish vaccinated with prototype B, the BP annotated in head kidney tissue were nitrogen compound metabolic process, macromolecule metabolic process, and cellular nitrogen compound metabolic process. While the MFs annotated were RNA polymerase II transcription regulatory region sequence-specific DNA binding, ubiquitin-like protein ligase activity, and ubiquitin-protein ligase activity (Figure 4). Furthermore, skin tissues of vaccinated fish with prototype B exhibited BPs associated with cell surface receptor signaling pathways involved in cell–cell signaling, organic acid metabolic process, and oxoacid metabolic process (Figure 5). In addition, the main MFs in skin samples of fish vaccinated with prototype B were hydrolase activity, iron–sulfur cluster binding, and metal cluster binding. Interestingly, fish vaccinated with prototype C showed the highest abundance of BPs and MFs differently modulated (Figure 4 and Figure 5). The BP widely annotated in the skin and head kidney tissue of fish vaccinated with prototype C was associated with cellular, metabolic, and biological regulation. While the MFs observed were hydrolase activity, iron–sulfur cluster binding, and metal cluster binding. Finally, Figure 6 shows the main BPs and MFs annotated for the sharing transcripts among the three prototypes. They highlighted BPs as a cellular process, biological regulation, and metabolic processes in the head kidney. Meanwhile, the skin samples of vaccinated salmons have annotated BPs as a cellular process, cellular component organization or biogenesis, and cellular component organization.

Similarly, the transcripts differently expressed among the experimental groups were annotated by Kegg pathway analyzes (Figure 7). The metabolic pathway exhibited the highest number of transcripts annotated in the three prototypes and both tissues. For instance, differently expressed transcripts of head kidney from fish vaccinated with prototype A, annotated for pathways related to cytokine–cytokine receptor interaction, cell adhesion molecules, and regulating signaling pathways of pluripotency of stem cells. While in skin tissue we observed transcripts associated with activated ribosome, focal adhesion, and endocytosis pathways. Furthermore, for prototype B, the head kidney showed high transcripts abundance of autophagy pathways, biosynthesis of secondary metabolites, and platelet activation. In addition, endocytosis, thermogenesis, and regulation of actin cytoskeleton pathways were annotated in skin samples of prototype B vaccinated fish. Notably, prototype C had a more significant number of differently expressed transcripts annotated. Among the pathways identified in the head kidney of fish vaccinated with prototype C were biosynthesis of secondary metabolites, regulation of actin cytoskeleton, thyroid hormone signaling pathway, and MAPK signaling pathway. Regarding skin tissues, prototype C showed transcripts associated with pathways related to the regulation of actin cytoskeleton, endocytosis, and osteoclast differentiation. Finally, the sharing transcripts expressed in head kidney tissue were annotated by the PI3K-Akt signaling pathway and phagosome pathway. While in skin samples, biosynthesis of secondary metabolites, PI3K-Akt signaling pathway, and MAPK signaling pathway was widely annotated among sharing transcripts.

### 3.4. RT-qPCR Evaluation of Genes Associated with C. Rogercresseyi Infestation 

Fish early infestation response was evaluated in vaccinated fish using a panel of genes associated with the *S. salar* response to sea lice infestation (Figure 8). The RT-qPCR analysis of head kidney tissue IgM and IgT were upregulated in fish vaccinated with prototypes A and B. In contrast, the Ferritin and heme oxygenase (HO) genes were downmodulated in vaccinated fish compared with the control group. Notably, skin fish vaccinated with prototype A exhibited upregulation of immune response genes such as IgM, COX-2, MHCII, TLR22, and ferritin gene compared with the control group and skin fish vaccinated with prototype C. Moreover, the skin fish vaccinated with prototype B exhibited upregulation of genes MCHII, TLR22, Ferritin, and hepcidin. Interestingly, all evaluated genes were down modulated in skin fish vaccinated with prototype C compared with the control group. The low expression of these genes in fish with prototype C could explain the low parasite load observed in this group that reduces the immune response activation.

### 3.5. Antigens’ Efficacy in Preventing Parasite Attachment

Parasite burden estimation was carried out in randomly selected three fish per tank, equivalent to nine individuals per fish/group. Notably, fish vaccinated with prototypes B and C exhibited an average load of four chalimus per fish, showing efficacies between 44 and 52% compared with the control group. Moreover, fish vaccinated with prototype A showed seven chalimus per fish or 24% of efficacy. While the control group exhibited an average of eight chalimus per fish (Figure 9).

## 4. Discussion

Reverse vaccinology is a strategy that changes the way of antigens selection for vaccine production. The massive description of the whole genome from a different organism, followed by the bioinformatic tools developed in the last years, allows the discovery of many candidates for vaccine antigens [8,9]. The recently published *C. rogercresseyi* genome [53] allows identifying proteins relevant to the sea lice infestation process and host immune response evasion. These proteins have a significant potential to be tested as antigens for sea lice control based on the fish vaccine strategy. Previously, our research group characterized cathepsin and peritrophin genes with high expression levels during *C. rogercresseyi* infestation stages [18,60]. Cathepsins are proteolytic enzymes that participate in the mechanisms of immune evasion and survival of parasites [61]. In sea lice, cathepsins are secreted as part of the parasite’s enzyme battery during pathogen–host interaction [19,61]. Furthermore, cathepsins are also involved in parasite biological processes such as molting, extracellular digestion, and embryogenesis [62,63,64]. In addition, the role of cathepsins in nutrient digestion, such as hemoglobin obtained from the host, has been reported in hematophagous parasites [13,14]. Peritrophins are chitin-binding proteins initially isolated from the peritrophic membrane in insect intestinal [24,25]. In insects, peritrophins have an essential role in host–parasite interaction and have been suggested as a protective role against ingested toxins, pathogens, and host immune components [22,23]. Additionally, in blood-sucking insects such as the malaria mosquito, *Anopheles gambiae* peritrophins limit free radical formation by heme groups captured from the hosts [24,27]. These cathepsin and peritrophin characteristics make them attractive targets for parasite control and have been used as antigens in prototype vaccines against mammalian and chicken parasites [31,65,66]. Using these proteins for vaccine development against aquaculture pathogens has been suggested [67,68]. In this study, we described the molecular profile of *S. salar* vaccinated with three vaccines prototypes formulated with peritrophin and cathepsin antigens during the early infestation process of *C. rogercresseyi*.

Host behavior, physiology, fitness, health, and metabolism are influenced by parasite infections [69,70,71,72]. To avoid the harmful effects of parasite infection, the aquaculture industry employs different strategies, such as the use of formulated diet, selective breeding, and vaccines, among others [73,74,75,76,77]. For sea lice control, the use of a functional diet has been reported, which includes immune-stimulating components that increase mucus proteins’ concentration and trigger a *Lepeophtheirus salmonis* burden reduction [78]. Additionally, a parasite burden reduction at different developmental stages using vaccine prototypes has been described [34]. Recently, *L. salmonis* proteins related to the midgut function have been evaluated as antigens, showing 31 and 35% of salmon protection. Moreover, a chimeric antigen design from different iron-transporters sea lice proteins (IPath^®^) evidenced the modulation of iron homeostasis and immune response genes in *S. salar*. Moreover, the IPath^®^ vaccines showed a 95% efficacy in *C. rogercresseyi* burden reduction [41,79]. This study evaluated the effects of three vaccine prototypes in sea lice infestation. It showed significant parasite bunder decreases in the fish group vaccinated with prototypes B and C concerning the control group. Furthermore, both prototypes, including a cathepsin antigen, have many modulated transcripts associated with metabolic and energy production processes. Moreover, the fish group vaccinated with prototype A did not significantly differ from the control group. In addition, this fish group exhibited a high number of modulated transcripts associated with developmental process and anatomical structure development, which could reduce the energy available in fish for an efficient response to sea lice infestation or fish performance.

The fish skin is the first defensive barrier against external stressors [80,81]. In sea lice, fixation has been reported as the activation of a process as tissue reconstitution and maturation for the re-epithelialization of fish skin [82,83,84]. In this study, the transcriptome analysis showed a high number of transcripts highly regulated in skin tissue of fish vaccinated with prototype C (combination of peritrophin and cathepsin) compared with fish skin from the group vaccinated with prototype A (peritrophin). The transcripts identified in vaccinated fish were found associated with cell repair and restructuring of the cytoskeleton. High enrichment of MARK signaling pathways, phagosomes, and platelet activation were exhibited. These results suggested a fish response to sea lice injuries and the recruitment of immune response components. Herein, myosin genes were upregulated in vaccinated fish, a gene previously reported to be associated with disease resistance and lymphocyte migration among endothelium of salmon infected with the salmon louse *L. salmonis* [38,85]. On the other hand, in mammals, it has been described that platelet is an essential component in the homeostasis phase during tissue repairing and participate in the production of attracting signals for cells involved in subsequent inflammation events [86,87]. Notably, *S. salar* treated with cortisol and infested with *L. salmonis* evidenced downregulation of platelet-derived proteins after 18 dpi [88]. A high number of transcripts was associated with platelet activation pathways in fish vaccinated with prototype B, suggesting an improving tissue regeneration process in fish exposed to the cathepsin antigen.

Moreover, our research group has observed *S. salar* infested with *C. rogercresseyi* modulation changes of immunoglobulins, MHCII, TLR22, and cyclooxygenase-2 (COX-2) [52,61,89]. Here, the vaccinated fish groups exhibited immune response activation compared with the control group. For instance, the expression of the COX-2 gene, associated with inflammatory response [61,90], increases in skin tissue of all treatments. While in the head kidney tissue of fish injected with prototypes B and C, they had significantly decreased expression of the COX-2 gene. In *Oncorhynchus gorbuscha*, an expression increase in MHC II was observed after 24 h of *L. salmonis* attachment [91]. Moreover, it has been suggested that MHCII is involved in the *S. salar* response to skin damage produced by *L. salmonis* [92,93]. In contrast, the MHC II gene expression levels of *S. salar* infested with *C. rogercresseyi* did not show significant differences in the not-infested control [94,95]. However, at 14 dpi of *C. rogercresseyi* infestation, the expression of the MHC II gene increase in the head kidney, and skin tissues of *S. salar* [52]. On the other hand, studies in *S. salar* report the activation of TLR22 in response to *C. rogercresseyi* [61,96]. In this study, an upregulation of TLR22 and MHCII *S. salar* skin of groups injected with prototypes A and B suggest a sea lice response improved the association with the vaccine’s prototypes.

Prototypes A and B showed IgT upregulated in the head kidney, while only prototype A had a significant increase in skin IgM expression. Furthermore, changes in immunoglobulins expression were observed in vaccinated fish. Additionally, immunoglobulins expression increase has been reported in fish vaccinated with sea IPath^®^ vaccine and TT-P0 antigen [35,41,79]. It is possible to suggest the activation of Ig genes due to fish vaccination with the proposed prototypes.

Metal ions play a critical role in biological processes as components of metabolism, enzymatic cofactors, and structural supports to mediate electron transport or immune response to pathogens infections [97,98,99,100]. Metal ion concentrations trigger immune responses during pathogens infections [99,101]. In addition, metal ions such as Zn and Fe are essential micronutrients for pathogen development obtained from the host. The regulation of host micronutrients is a defense strategy known as nutritional immunity [42,102,103]. This strategy has been described in *S. salar* during *C. rogercresseyi* infestation [42]. Additionally, expression changes of iron homeostasis-related genes in *S. salar* exposed to *L. salmonis* and *C. rogercresseyi* have been described. Notably, genes related to heme degradation were upregulated in *S. salar* infected with *C. rogercresseyi* [42]. Furthermore, upregulation of iron homeostasis-related genes during *C. rogercresseyi* infestation in susceptible species *S. salar*, compared with the resistant Coho salmon has been described [50]. Furthermore, our research group has evidenced a sea lice load reduction in *S. salar* vaccinated with the IPath^®^ vaccine, a chimeric antigen with iron transport function [41]. Fish vaccinated with IPath^®^ showed a 96% reduction in adults’ *C. rogercresseyi* burden [41]. Notably, fish injected with prototype B, with a low parasite burden, showed an upregulation of heme group biosynthesis genes (ALAs, ALAd). Additionally, the hepcidin gene was upregulated in prototype B vaccinated fish compared with the other treatments. In skin tissue, prototypes A and B exhibit high expression levels of the Ferritin gene. Herein, it is possible to suggest the effects of the antigens in *S. salar* iron regulation during a sea lice infestation.

## 5. Conclusions

This study analyzed the transcriptomic modulation of *S. salar* immunized with three vaccine prototypes during the early infection of *C. rogercresseyi*. Prototype-dependent expression patterns were observed. Vaccine prototypes B and C, both with cathepsin antigen, exhibited a significant decrease in the number of *C. rogercresseyi* attached. Furthermore, these fish showed a high modulation of metabolic processes and metal ions binding. Finally, vaccinated fish with three vaccine prototypes showed modulation changes in biological processes such as biological regulation, cellular metabolic processes, and energy production, which may be fundamental to the early stages of *C. rogercresseyi* fish response.

## Figures and Tables

**Figure 1 vaccines-10-01063-f001:**
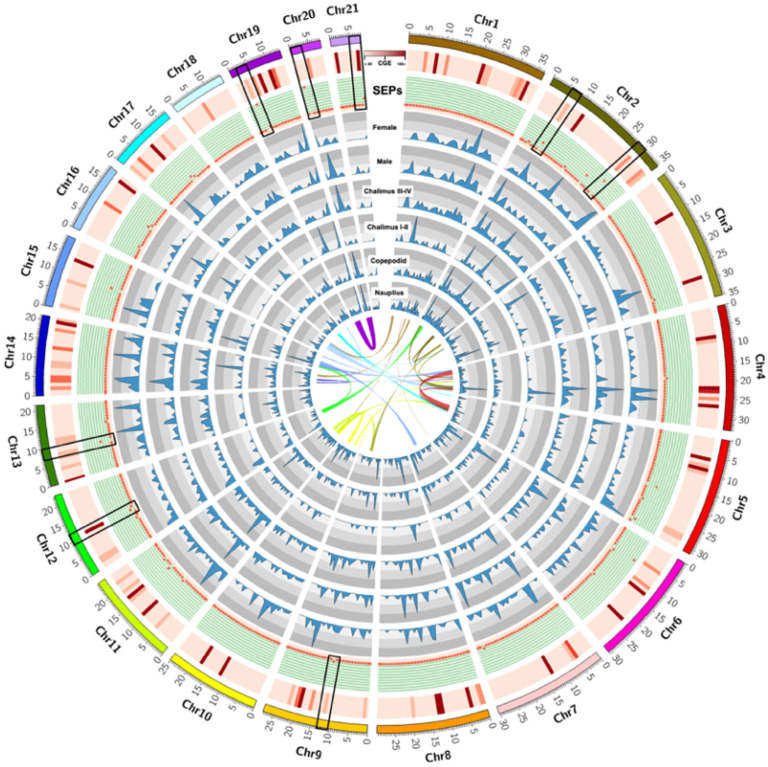
*Caligus rogercresseyi* transcriptome showing threshold areas for differential gene expression activity during the ontogeny. The graph tracks identify graph regions that fall between 10–20 K reads of coverage values of each chromosome. CGE index: heatmap in red showed the differences in expression variation among sea lice developmental stages. SEPs: red dots represent the secretome-related genes on each differential chromosome region. Blue profiling indicates the transcriptome coverage for each ontogenetic stage. Syntenic relationships of gene blocks among *C. rogercresseyi* pseudochromosomes are also displayed.

**Figure 2 vaccines-10-01063-f002:**
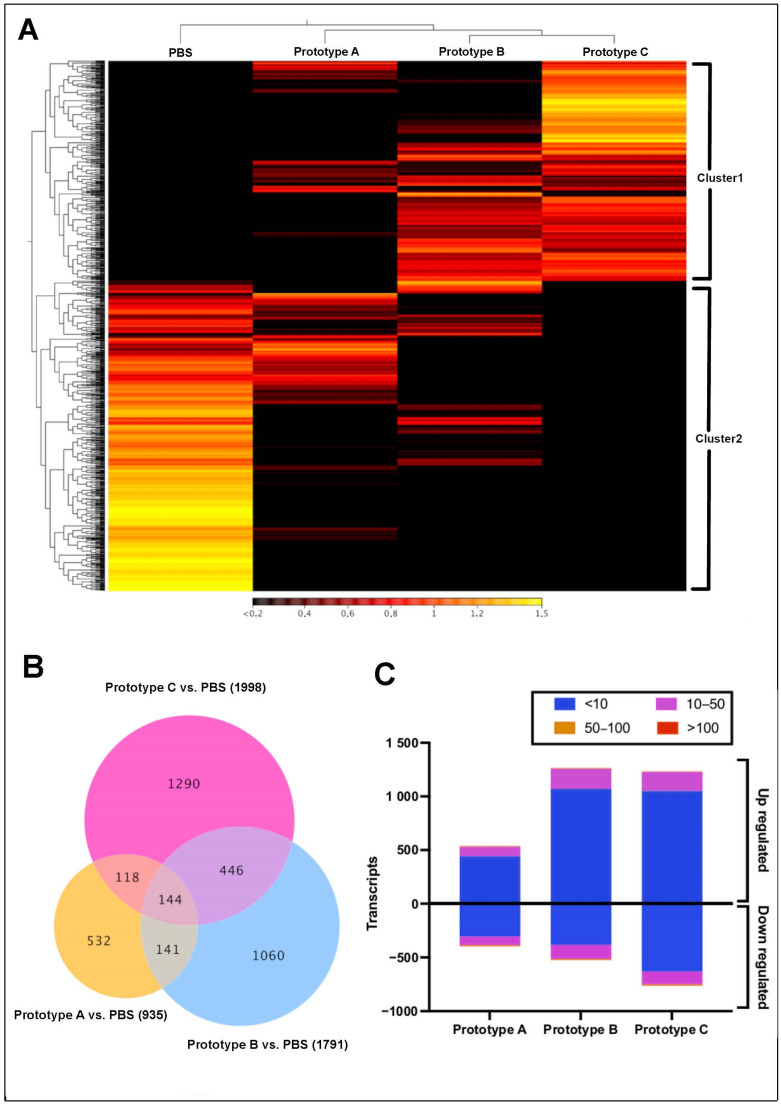
Transcriptomic profiling of *S. salar* head kidney tissue under experimental conditions. (**A**) Heatmap showing expression values expressed in TPM. (**B**) Venn diagram of contigs differentially expressed respect with the control group (false discovery rate (FDR) 4 and *p*-value 0.05). The number in parentheses represents the total contigs for each treatment. (**C**) Distribution of fold change values in each experimental group.

**Figure 3 vaccines-10-01063-f003:**
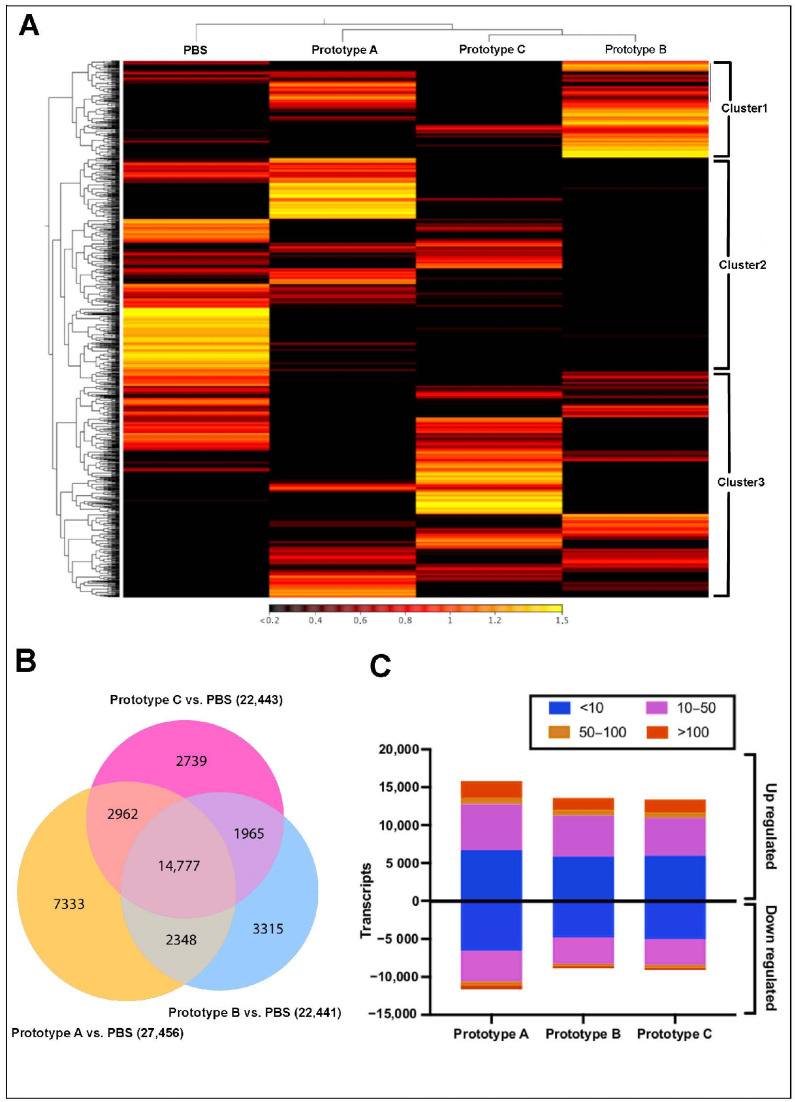
Transcriptomic profiling of vaccinated *S. salar* skin tissue. (**A**) Heatmap showing expression values expressed in TPM. (**B**) Venn diagram of contigs differentially expressed respect with the control group (FDR 4 and *p*-value 0.05). The number in parentheses represents the total contigs for each treatment. (**C**) Distribution of fold change values in each experimental group.

**Figure 4 vaccines-10-01063-f004:**
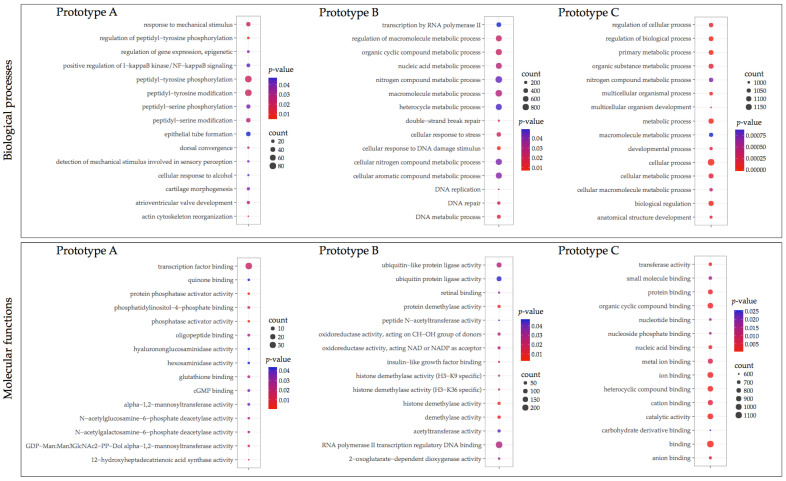
Gene ontology enrichment of differentially expressed transcripts in the head kidney of vaccinated fish groups. The size of the circles indicates the number of transcripts that enriched the GO term. Colors indicate the *p*-value.

**Figure 5 vaccines-10-01063-f005:**
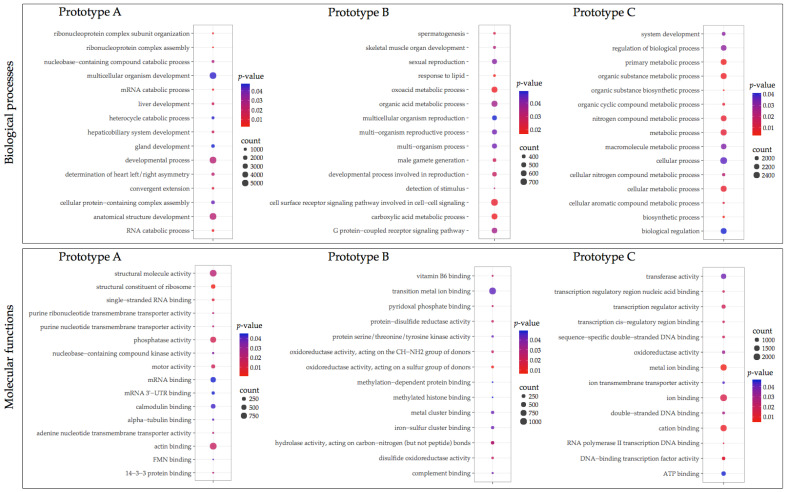
Gene ontology enrichment analysis of differentially expressed transcripts in the skin of vaccinated fish groups. The size of the circles indicates the number of transcripts that enriched the GO term. Colors indicate the *p*-value.

**Figure 6 vaccines-10-01063-f006:**
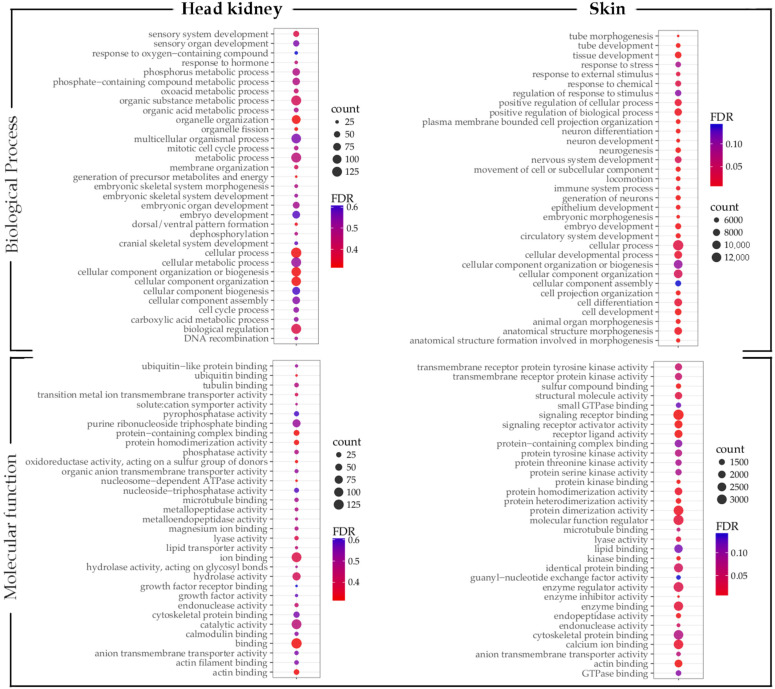
GO enrichment analysis of shared expressed transcripts in vaccinated fish groups. The size of the circles indicates the number of transcripts that enriched the GO term. Colors indicate the *p*-value.

**Figure 7 vaccines-10-01063-f007:**
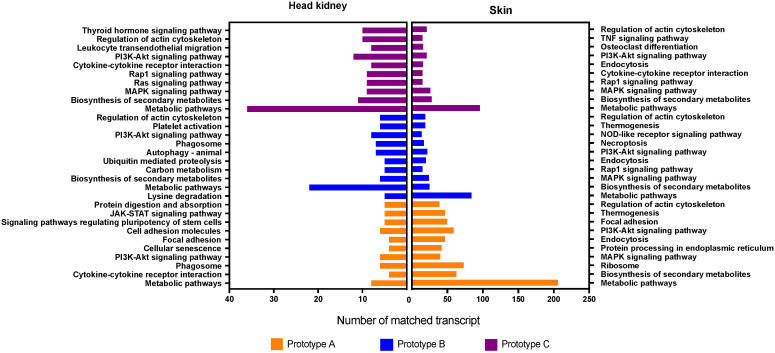
KEGG pathway enrichment analysis of differential expressed contigs. Bars represent the enriched contigs count for the pathway.

**Figure 8 vaccines-10-01063-f008:**
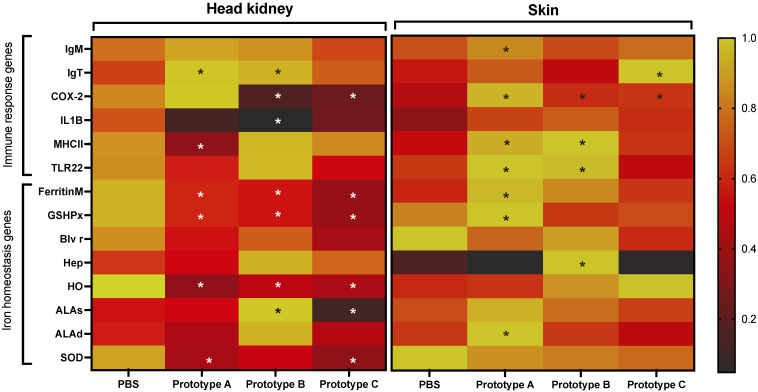
RT-qPCR analisys of *S. salar* response *C. rogercresseyi* infestation. The heatmap shows the relative expression levels. Asterisks showed significant differences concerning the control group (PBS: phosphate-buffered saline) at *p*-value < 0.05, black asterisks upregulation, white asterisks downregulation.

**Figure 9 vaccines-10-01063-f009:**
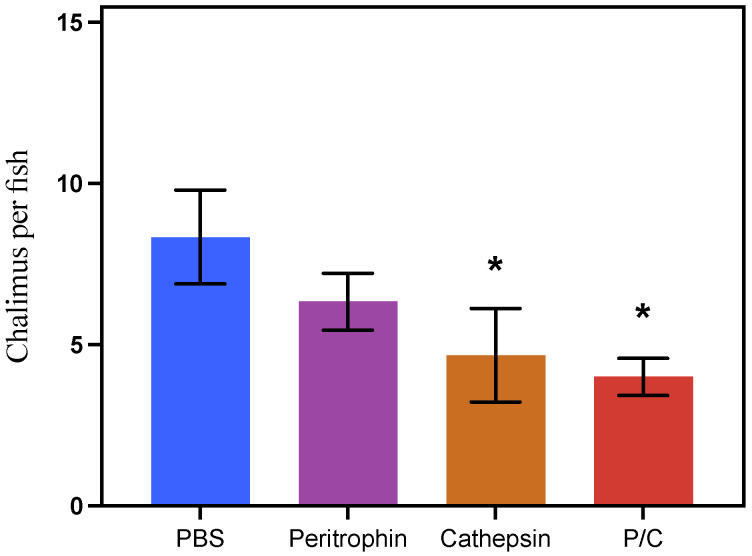
Chalimus I burden per fish in each experimental group. Asterisks showed significant differences concerning the control group (PBS: phosphate-buffered saline) (*p*-value < 0.05).

**Table 1 vaccines-10-01063-t001:** Annotation of the differentially expressed contigs in head kidney tissue of vaccinated *S. salar*. Heatmap indicates the levels of regulation for each gene with blue at the top and red at the bottom. Red arrows indicate gene downregulation, and green arrows indicate upregulation.

Prototype	Feature ID	Fold Change	Lowest E-Value	Annotation
A	contig_0004466	↑4960.596	2	RNA-binding protein 44 [*Danio rerio*]
	contig_0004597	↑25.248	4	Manganese-dependent ADP-ribose/CDP-alcohol diphosphatase [*Danio rerio*]
	contig_0020187	↑182.493	1	Huntingtin [*Takifugu rubripes*]
	contig_0025250	↑21.707	6	Protein Jumonji [*Danio rerio*]
	contig_0034576	↑24.093	5	Transmembrane protein 208 [*Danio rerio*]
	contig_0046089	↑18.726	4	Glucocorticoid receptor [*Oncorhynchus mykiss*]
	contig_0053965	↑18.575	0.7	Protein broad-minded [*Danio rerio*]
	contig_0055996	↓−32.997	0.4	Heparan-sulfate 6-O-sulfotransferase 1-B [*Danio rerio*]
	contig_0056158	↑19.593	5	Neurexin-1b [*Danio rerio*]
	contig_0071724	↑29.905	0.6	Disks large homolog 1[*Danio rerio*]
	contig_0071725	↓−43.599	0.3	Rab-like protein 3 [*Danio rerio*]
	contig_0080432	↑57.479	1 × 10^−5^	POC1 centriolar protein homolog A [*Danio rerio*]
	contig_0082059	↓−19.149	0.2	Protein tilB homolog [*Danio rerio*]
	contig_0129067	↓−20.274	0.6	Nuclear receptor ROR-alpha A [*Danio rerio*]
	contig_0151557	↓−27.038	0.5	Cyclic nucleotide-gated cation channel [*Ictalurus punctatus*]
	contig_0163106	↓−23.455	7 × 10^−144^	Trimeric intracellular cation channel type A [*Danio rerio*]
	contig_0203229	↓−19.214	1	Cell surface hyaluronidase [*Danio rerio*]
	contig_0278209	↓−19.172	2 × 10^−7^	Striated muscle preferentially expressed protein kinase [*Danio rerio*]
	contig_0322455	↓−32.997	0.9	Creatine kinase, testis isozyme [*Oncorhynchus mykiss*]
	contig_0425609	↑75.365	0.6	U11/U12 small nuclear ribonucleoprotein 35 kDa protein [*Danio rerio*]
	contig_0685545	↓−47.84	7 × 10^−46^	Myosin heavy chain, fast skeletal muscle [*Cyprinus carpio*]
B	contig_0004420	↓−36.35	0.7	Sodium channel protein type 4 subunit alpha A [*Takifugu rubripes*]
	contig_0014672	↓−36.35	0.07	Centrosomal protein kizuna [*Danio rerio*]
	contig_0015979	↓−25.976	0.5	PR domain zinc finger protein [*Danio rerio*]
	contig_0050531	↑195.503	0.8	DENN domain-containing protein 11 [*Danio rerio*]
	contig_0059800	↓−29.103	4	Heat shock 70 kDa protein 1 [*Oryzias latipes*]
	contig_0071672	↓−28.198	0.5	Transmembrane protein 53 [*Danio rerio*]
	contig_0075736	↓24.574	1	Ubiquitin carboxyl-terminal hydrolase 16 [*Danio rerio*]
	contig_0078326	↓−28.198	0.03	Protein tweety homolog 3 [*Danio rerio*]
	contig_0091637	↑34.302	5	Rab9 effector protein with kelch motifs [*Danio rerio*]
	contig_0096326	↑50.499	3	Transmembrane protein 116 [*Danio rerio*]
	contig_0099678	↑210.084	2	RNA-binding protein PNO1 [*Oryzias latipes*]
	contig_0111985	↑45.465	1	Ankyrin repeat and IBR domain-containing protein 1 [*Danio rerio*]
	contig_0122208	↓−20.951	4 × 10^−94^	Putative deoxyribonuclease tatdn3 [*Danio rerio*]
	contig_0133151	↑527.634	0.2	E3 ubiquitin-protein ligase MYCBP2 [*Danio rerio*]
	contig_0158353	↑31.508	2	Interferon regulatory factor 2-binding protein 2-B [*Danio rerio*]
	contig_0252619	↑48.853	0.06	Polycomb protein suz12-B [*Danio rerio*]
	contig_0328706	↓−47.906	5	E3 ubiquitin-protein ligase MYCBP2 [*Danio rerio*]
	contig_0389545	↑137.058	7	NADH-ubiquinone oxidoreductase chain 1 [*Gadus morhua*]
	contig_0473006	↓−84.357	5	Guanine nucleotide exchange protein smcr8a [*Danio rerio*]
C	contig_0004540	↑28.043	2	Pyridoxal-dependent decarboxylase domain-containing protein 1 [*Danio rerio*]
	contig_0022126	↓−34.603	0.2	CD166 antigen homolog A [*Danio rerio*]
	contig_0047184	↓−19.98	0.2	Calcium/calmodulin-dependent protein kinase type II delta 2 chain [*Danio rerio*]
	contig_0055363	↑18.009	3	V(D)J recombination-activating protein 2 [*Oncorhynchus mykiss*]
	contig_0094127	↑20.876	5	Poly(A)-specific ribonuclease PARN [*Danio rerio*]
	contig_0098235	↑18.725	1	Neurexin-1b [*Danio rerio*]
	contig_0104613	↑18.725	0.2	Src kinase-associated phosphoprotein 1 [*Takifugu rubripes*]
	contig_0105909	↑21.592	0.2	RING finger protein 145 [*Danio rerio*]
	contig_0112666	↓−21.083	1	Ribonucleoside-diphosphate reductase large subunit [*Danio rerio*]
	contig_0117074	↓−18.745	0.6	Sodium channel protein type 4 subunit alpha B [*Takifugu rubripes*]
	contig_0134124	↓−19.98	0.8	E3 ubiquitin-protein ligase TRIP12 [*Danio rerio*]
	contig_0137098	↓−18.878	1	Threonine synthase-like 2 [*Danio rerio*]
	contig_0145112	↓−21.083	1	Sodium channel protein type 4 subunit alpha B [*Danio rerio*]
	contig_0158020	↑20.159	1 × 10^−7^	Contactin-5 [*Danio rerio*]
	contig_0158413	↓−21.083	5	Treslin [*Danio rerio*]
	contig_0292034	↑26.614	10	Alpha-protein kinase 2 [*Danio rerio*]
	contig_0402943	↑54.564	4	RAB11-binding protein RELCH homolog [*Danio rerio*]
	contig_0430511	↓−18.878	2 × 10^−18^	Creatine kinase, testis isozyme [*Oncorhynchus mykiss*]
	contig_0540911	↓−69.586	1	N-acetylneuraminate 9-O-acetyltransferase [*Danio rerio*]

**Table 2 vaccines-10-01063-t002:** Annotation of the differentially expressed contigs in skin tissue of vaccinated *S. salar*. Heatmap indicates the levels of regulation for each gene with blue at the top and red at the bottom. Red arrows indicate gene downregulation, and green arrows indicate upregulation.

Prototype	Feature ID	Fold Change	Lowest E-Value	Annotation
A	contig_0000167	↓−606.592	8 × 10^−23^	Formin-like protein 3 [*Danio rerio*]
	contig_0008746	↓−527.023	0.4	Serine/threonine-protein kinase N2 [*Danio rerio*]
	contig_0012767	↓−606.592	5	Transmembrane protein 198-B [*Danio rerio*]
	contig_0039965	↓−575.988	1	Acylglycerol kinase, mitochondrial [*Danio rerio*]
	contig_0046974	↓−594.350	2	Xenotropic and polytropic retrovirus receptor 1 homolog [*Danio rerio*]
	contig_0058658	↓−563.747	2 × 10^−11^	Melanocortin-2 receptor accessory protein 2B [*Danio rerio*]
	contig_0061338	↓−575.988	2	Hemoglobin subunit beta [*Thunnus thynnus*]
	contig_0083287	↓−588.230	2	Neurexin-3b [*Danio rerio*]
	contig_0096893	↑40,425.799	7 × 10^−131^	Elongation factor 1-alpha [*Danio rerio*]
	contig_0097840	↓−545.385	0	V(D) J recombination-activating protein 1 [*Oncorhynchus mykis*]
	contig_0143302	↓−533.143	0.6	Polycystin-2 [*Oryzias latipes*]
	contig_0185421	↑45,501.610	6	Estrogen receptor [*Ictalurus punctatus*]
	contig_0321861	↑227,716.450	0	Cytochromec oxidase subunit 1 [*Danio rerio*]
	contig_0552888	↑49,171.662	5 × 10^−3^	Vitellogenin-1 [*Fundulus heteroclitus*]
	contig_0571513	↑33,289.567	9	Lysine-specific demethylase phf2 [*Dicentrarchus labrax*]
	contig_0571697	↑36,111.859	1	Thrombospondin type-1 domain-containing protein 7A [*Danio rerio*]
	contig_0571705	↑28,449.821	2 × 10^−24^	NADH-ubiquinone oxidoreductase chain 5 [*Carassius auratus*]
	contig_0571764	↑39,320.471	2 × 10^−3^	Dynein heavy chain (Fragment) [*Oncorhynchus mykiss*]
	contig_0571817	↑58,078.512	3	Unconventional myosin-IXAa [*Danio rerio*]
	contig_0571876	↑30,252.653	9	Sodium channel protein type 4 subunit alpha B [*Danio rerio*]
B	contig_0016642	↓−38.578	6	Collagen alpha-1(XXVII) chain B [*Danio rerio*]
	contig_0020177	↓−34.554	8	Xin actin-binding repeat-containing protein 1 [*Danio rerio*]
	contig_0030309	↓−31.679	2 × 10^−13^	Palmitoyltransferase [*Danio rerio*]
	contig_0035782	↓−36.278	0.3	CCR4-NOT transcription complex subunit 10 [*Danio rerio*]
	contig_0035986	↓−34.554	2 × 10^−74^	Complement component C9 (Fragment) [*Oncorhynchus mykiss*]
	contig_0055961	↓−31.104	−0.8	3-hydroxybutyrate dehydrogenase type 2 [*Danio rerio*]
	contig_0062345	↑2392.329	0.2	Neuroblastoma-amplified sequence [*Danio rerio*]
	contig_0066929	↓−38.578	0.5	Disks large-associated protein 1 [*Danio rerio*]
	contig_0075266	↓−35.128	0.1	Eukaryotic translation initiation factor [*Danio rerio*]
	contig_0102002	↓−29.954	0.6	Presequence protease, mitochondrial [*Danio rerio*]
	contig_0164965	↑2915.165	0.03	Methionine-tRNA ligase, mitochondrial [*Takifugu rubripes*]
	contig_0236931	↑2900.227	9	CCR4-NOT transcription complex subunit 1 [Danio rerio]
	contig_0287222	↓−34.554	0.9	Striatin interacting protein 1 homolog [*Danio rerio*]
	contig_0359850	↑2213.070	0.3	Gonadotropin subunit beta-2 [*Anguila anguilla*]
	contig_0454328	↑2168.256	0.9	Paired box protein Pax-6 [*Oryzias latipes*]
	contig_0492154	↑2332.576	7 × 10^−10^	WAP, Kazal, immunoglobulin, Kunitz and NTR domain-containing protein [*Danio rerio*]
	contig_0557190	↑2511.834	1	Neuron navigator 3 [*Danio rerio*]
	contig_0572018	↑2646.278	1	2-oxoglutarate and iron-dependent oxygenase domain-containing protein 2 [*Danio rerio*]
	contig_0581749	↑2362.452	4 × 10^−15^	Creatine kinase, testisisozyme [*Oncorhynchus mykiss*]
	contig_0635620	↑2930.103	6	Lactosylceramide 1,3-Nacetyl-beta-D-glucosaminyl transterase B [*Danio rerio*]
C	contig_0002190	↓−27.536	2	Somatolactin [*Cyclopterus lumpus*]
	contig_0010441	↓−28.054	1	Ubiquinol-cytochrome-c reductase complex assembly factor 3 [*Danio rerio*]
	contig_0013985	↓−28.572	4	Insulin-like growth factor 1, adult form [*Cyprinus carpio*]
	contig_0038889	↓−33.756	8	Plexin-A4 [*Danio rerio*]
	contig_0042915	↓−28.572	2	SUMO-specific isopeptidase USPL1 [*Danio rerio*]
	contig_0052507	↓−35.830	1 × 10^−4^	Pyridine nucleotide-disulfide oxidoreductase domain-containing protein [*Danio reric*]
	contig_0068071	↓−29.091	1	Probable serpin E3 [*Danio rerio*]
	contig_0085140	↓−25.980	0.5	CCR4-NOT transcription complex subunit 9 [*Danio rerio*]
	contig_0130362	↑1779.731	1	NADH-ubiquinone oxidoreductase chain 2 [*Salmo solar*]
	contig_0141833	↓−30.128	2	Neuropilin-1a [*Danio rerio*]
	contig_0153864	↓−29.609	5	Polyribonucleotide 5′-hydroxyl-kinase Clp1 [*Danio rerio*]
	contig_0264422	↑1765.159	2	Semaphorin-3D [*Danio rerio*]
	contig_0415197	↑1677.727	0.3	Vascular endothelial growth factor receptor 2 [*Danio rerio*]
	contig_0434444	↑1619.438	0.2	DNA excision repair protein ERCC-6-like [*Danio rerio*]
	contig_0491119	↑1692.299	1	Glucagon family neuropeptides [*Clarios macrocephalus*]
	contig_0531987	↑2260.609	2 × 10^−46^	Sarcoplasmic/endoplasmic reticulum calcium ATPase 1 [*Makaira nigricans*]
	contig_0561562	↑1590.294	0.5	Membrane progestin receptor alpha [*Cynoscion nebulosus*]
	contig_0572520	↑1983.740	3	Protein jagunal homolog 1A [*Danio rerio*]
	contig_0577243	↑1910.879	2	Galectin-3-binding protein A [*Donio rerio*]
	contig_0596536	↑1867.163	0.4	POU domain, class 6, transcription factor 1 [*Danio rerio*]

## Data Availability

This study did not report any data.

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
