# Peer review of "Exploring Sea Lice Vaccines against Early Stages of Infestation in Atlantic Salmon (Salmo salar)"

_vaccines, 2022, doi:10.3390/vaccines10071063_

Round 1
Reviewer 1 Report
The manuscript with ID (vaccines-1717345) by Casuso and coauthors has tested the efficacy of three prototype vaccines during early stages of sea lice infestation in Atlantic salmon. In this regard, authors have studied gene expression analysis to evaluate the effects of these vaccines on modulation of genes involved in immune responses, iron transport, and stress responses. The study is interesting; however, several revisions to be attempted prior considering this manuscript for publication in Vaccines.
Questions: -
As an ectoparasite, the sea lice Caligus rogercresseyi infests salmon skin, why authors have evaluated the gene expression analysis in head kidney tissue? I found that gene expression analysis in the integument is accurately defined only. Please give appropriate explanation.
Minor revisions: -
There are several revisions in the manuscript that should be revised carefully by authors. Please see the attached PDF file.
Lines 19-20: “The vaccines prototypes showed efficacies between 24 to 52% compared with the control group”. This sentence is too general- Authors should specify the efficacy of each one.
Line 33: viral
Line 34: in aquaculture
Line 38: for parasite control. Remove “treatment”
Line 51: Our recently published study demonstrated that IPath® vaccine,
Line 55: the sea louse, Caligus
Reference section
There are several revisions in reference section
- All Latin names should be written italicized.
- All capital initials in all references should be revised.
- Journal names should be revised as in the following references
- You should be consistent either write the full name of journals or abbreviate them

Author Response
Dear reviewer,
We do appreciate your suggestions to improve our manuscript. Please find below the responses to every question and/or comment that you have made. Furthermore, the corrections and suggested changes to the manuscript were made using “Track Changes” to facilitate its visualization. Also, the manuscript is highlighted in different color for each reviewer’s comment. Furthermore, as suggest reviewer 1 the English was improving.
Yours Sincerely,
The corresponding author
Reviewer 1 (green highlighted)
As an ectoparasite, the sea lice Caligus rogercresseyi infests salmon skin, why authors have evaluated the gene expression analysis in head kidney tissue? R. Thanks for the question. In teleosts, head kidney is an important hematopoietic-lymphoid and endocrine organ. It has been reported a high modulation change of immune-related genes in Atlantic salmon head kidney during sea lice infestation (references 41, 42, 50, 53, 80, 90, 96). For instance, has been reported an upregulation of TLR21 and TLR22a2 in head kidney of Atlantic salmon after 7 days of C. rogercreseyi infestation (53). Also, changes in Th2 genes have been observed in head kidney tissue at 14 dpi with C. rogercresseyi (53). Due to the goal of this study was to determine the effects of vaccine prototypes' in Atlantic salmon immune response, for us, the transcriptomic analysis of head kidney tissue is relevant to understanding the immune modulation effects of vaccines prototypes.
I found that gene expression analysis in the integument is accurately defined only. Please give appropriate explanation. R. Thanks for the comments. The methodology for expression analysis was re-writing. Also, we added a supplementary Table S1 with primers information. (lines 188, 290-293, 196-199, 202)
Minor revisions:
There are several revisions in the manuscript that should be revised carefully by authors. Please see the attached PDF file. R. Thanks for the comments and your suggestion in the PDF file. Revisions included in the PDF file were made as has been suggested by the reviewer. (lines 3, 18-21, 33, 34, 38, 67, 72, 134, 254, 260, 358)
Lines 19-20: “The vaccines prototypes showed efficacies between 24 to 52% compared with the control group”. This sentence is too general- Authors should specify the efficacy of each one. R. Thanks for the comment. The sentence was rewritten, indicating the efficacy for each vaccine prototype (lines 18-21)
Line 33: viral R. Thanks for the comment, the sentence was change as the reviewer suggest. (line 33)
Line 34: in aquaculture. R. Thanks for the suggestion, the sentence was change. (line 34)
Line 38: for parasite control. Remove “treatment”. R. Thanks for the comment, the sentence was change as the reviewer suggest. (line 38)
Line 51: Our recently published study demonstrated that IPath® vaccine. R. Thanks for the comment, the sentence was change as the reviewer suggest. (line 67)
Line 55: the sea louse, Caligus. R. Thanks for the suggestion, the sentence was change. (line 72)
Reference section
- All Latin names should be written italicized. R. Thanks for the comment. Latin names were italicized as the reviewer suggest.
- All capital initials in all references should be revised. R. Thanks for the comment, changes were made as the reviewer suggest.
- Journal names should be revised as in the following references R. Thanks for the comment, the journal names were formatted to Capitalize for Each Word.
- You should be consistent either write the full name of journals or abbreviate them. R. Thanks for the suggestion, we standardized the references section using the full name of journals.

Reviewer 2 Report
The manuscript “Exploring sea lice vaccines against early stages of infestation in Atlantic salmon” from Casuso et al. is about experimentals trials of sea lice vaccines atlantic salmon. The in vivo findings are underlined with NGS data and qPCR. The presented work is in a good quality, the study design is well prepared and presentation and discussion of the results is convincing. With the presented work an important topic in aquaculture is emphasized.
General comments:
I would advice the authors to add information of cathepsin and peritrophin in the introduction for an easier access to the topic.
Unfortunately, the data are based on expression analysis by NGS or qPCR only. Proteom and / or serology would be a great benefit.
Besides all the fancy figures on NGS analysis, the authors should provide a figure about in vivo data.
Author Response
Dear reviewer,
We do appreciate your suggestions to improve our manuscript. Please find below the responses to every question and/or comment that you have made. Furthermore, the corrections and suggested changes to the manuscript were made using “Track Changes” to facilitate its visualization. Also, the manuscript is highlighted in different color for each reviewer’s comment. Furthermore, as suggest reviewer 1 the English was improving.
Yours Sincerely,
The corresponding author
Reviewer 2 (blue highlighted)
General comments:
I would advice the authors to add information of cathepsin and peritrophin in the introduction for an easier access to the topic. R. Thanks for the comment. In the introduction section was include more information about both proteins, for better understanding. (lines 42-58).
Unfortunately, the data are based on expression analysis by NGS or qPCR only. Proteome and / or serology would be a great benefit. R. Thanks for the comment. For this study did not was considered in the experimental design samples for proteomic of serology analysis, thus is not possible include results associated. For future studies we will have mind you suggestion.
Besides all the fancy figures on NGS analysis, the authors should provide a figure about in vivo data. R. Thanks for the comment. In response to the reviewer's suggestion, we add the information as a supplementary Figure S1. (line 131)
Round 2
Reviewer 1 Report
The authors have properly responded to the points raised by the anonymous reviewer
Author Response
Dear reviewer,
We do appreciate your suggestions to improve our manuscript.